# Genome-Wide Identification and Expression Profiling of Heat Shock Protein 20 Gene Family in *Sorbus pohuashanensis* (Hance) Hedl under Abiotic Stress

**DOI:** 10.3390/genes13122241

**Published:** 2022-11-29

**Authors:** Xiangyu Qi, Zexin Di, Yuyan Li, Zeren Zhang, Miaomiao Guo, Boqiang Tong, Yizeng Lu, Yan Zhang, Jian Zheng

**Affiliations:** 1School of Landscape Architecture, Beijing University of Agriculture, Beijing 102206, China; 2Shandong Provincial Center of Forest and Grass Germplasm Resources, Jinan 250102, China

**Keywords:** small heat shock protein, abiotic stress, gene family, expression analysis, *Sorbus pohuashanensis*

## Abstract

Small heat shock proteins (HSP20s) are a significant factor in plant growth and development in response to abiotic stress. In this study, we investigated the role of HSP20s’ response to the heat stress of *Sorbus pohuashanensis* introduced into low-altitude areas. The *HSP20* gene family was identified based on the genome-wide data of *S. pohuashanensis*, and the expression patterns of tissue specificity and the response to abiotic stresses were evaluated. Finally, we identified 38 *HSP20* genes that were distributed on 16 chromosomes. Phylogenetic analysis of HSP20s showed that the closest genetic relationship to *S. pohuashanensis* (SpHSP20s) is *Malus domestica*, followed by *Populus trichocarpa* and *Arabidopsis thaliana*. According to phylogenetic analysis and subcellular localization prediction, the 38 SpHSP20s belonged to 10 subfamilies. Analysis of the gene structure and conserved motifs indicated that *HSP20* gene family members are relatively conserved. Synteny analysis showed that the expansion of the *SpHSP20* gene family was mainly caused by segmental duplication. In addition, many cis-acting elements connected with growth and development, hormones, and stress responsiveness were found in the *SpHSP20* promoter region. Analysis of expression patterns showed that these genes were closely related to high temperature, drought, salt, growth, and developmental processes. These results provide information and a theoretical basis for the exploration of *HSP20* gene family resources, as well as the domestication and genetic improvement of *S. pohuashanensis*.

## 1. Introduction

As sessile organisms, plants are often subjected to various types of adversity, such as drought, heat, salt, cold, heavy metals, diseases, pests, and damage caused by human activity and other animal activities [1,2,3]. As the global temperature has increased in recent years, the damage caused to plants by heat stress has become more serious [4], often causing osmotic and oxidative stress and seriously affecting the growth, development, and natural distribution of plants [5]. To adapt to various adverse growth environments, plants have developed complex self-protection mechanisms throughout their long-term evolution [6,7] Heat shock proteins (HSPs), members of the heat stress-related protein family, mediate maintaining the maintenance of cell homeostasis [8] and are involved in embryo development, seed germination, fruit ripening, and other growth and development processes [9].

Heat and other stressors (such as salt, drought, cold, oxidative stress, heavy metals, and disease) induce the production of HSPs [6]. Under stress conditions, HSPs maintain the stability of proteins and membranes by participating in protein folding, transport, repair, degradation, and signal transduction, reducing the damage to plants [10,11]. According to molecular weight (MW) and sequence homology, HSPs are divided into five families: HSP100s, HSP90s, HSP70s, HSP60s, and HSP20s [12]. The molecular weight of HSP20s, also known as small heat shock proteins (sHSPs), is between 12 and 45 kilodaltons (kDa). Studies have shown that HSP20s account for the largest proportion of HSPs in plants [13] and are the most plentiful protein products under heat stress in many higher plants [14,15]. One of the characteristics of HSP20s is that they avoid protein denaturation as ATP-independent molecular chaperones by forming oligomeric protein complexes with 9–50 subunits (200–800 kDa) [16].

Another characteristic of HSP20s is that they have a relatively conserved structure that consists of three parts: the variable N-terminal, the relatively conserved C-terminal (alpha-crystalline domain, ACD), and the C-terminal extension region. The conserved C-terminal domain, also known as the HSP20 domain, contains approximately 80–100 amino acids and is composed of seven or eight contrarily parallel beta chains that fold into a compact sandwich structure. It can be divided into two conserved regions (CRs): one is CRI (β2, β3, β4, β5) at the N-terminal and the other is CRII (β7, β8, β9) at the C-terminal. These two regions are separated by a hydrophobic β6-loop with a variable length [16,17]. The N-terminus is involved in the regulation of oligomerization and substrate binding. The ACD interacts with substrates, and the C-terminus contains amino acid motifs that maintain organelle specificity and promote oligomerization and chaperoning [18]. The upstream promoter regions of the *HSP20*-encoding sequences usually contain multiple repeated 5′-nGAAnnTTCnnGAAn-3′ sequences (heat shock element, HSE), which are identified and bound by specific heat shock transcription factors (Hsfs) [13].

HSP20s can be divided into 12 subfamilies based on the subcellular location and homology of the sequence, among which seven subfamilies (CI, CII, CIII, CIV, CV, CVI, and CVII) are located in the cytoplasm or nucleus, and two subfamilies (MI, MII) are located in the mitochondria. The other three subfamilies (ER, P, and Po) are located in the endoplasmic reticulum, plastid, and peroxisome, respectively [15,19]. The number of HSP20s in plants is much greater than that in prokaryotes and animals [12]. For example, 39, 42, 51, 35, and 19, HSP20s have been identified in rice (*Oryza sativa*) [15], tomato (*Solanum lycopersicum*) [20], soybean (*Glycine max*) [21], pepper (*Capsicum annuum*) [22], and *A. thaliana* [23], respectively, but only 10 HSP20s have been identified in humans [24].

One of the main effects of HSP20s is to regulate the plant response to heat stress. Overexpression of *HSP20* genes in plants has been shown to improve their tolerance to heat and oxidative stress through protection of photosystem II and an increase in the activity of peroxidase [25,26]. HSP20s are also involved in other functions in plants, such as their response to abiotic stresses, growth, and development [27]. For example, HSP20s enhances seed germination under salt stress and plant tolerance to cold stress by increasing the activity of superoxide dismutase or catalase. It also regulates the response of plants to salt stress through the abscisic acid signaling pathway [28]. It has been reported that HSP20s are involved in the transformation of chloroplasts into pigment mother cells, which promotes fruit color change [9]. Moreover, recent studies have shown that HSP20s mediate plant immune processes [29,30].

*S. pohuashanensis* is a deciduous tree belonging to the *Sorbus* spp. of Rosaceae, and its leaves, flowers, and fruits are of high ornamental value, making it an important ornamental tree species in gardens. However, because *S. pohuashanensis* naturally grows in high altitudes (800–2000 m), when it is introduced to low-altitude areas, it is often affected by heat stress in the summer, which causes wilting and sunburn that seriously affects its ornamental value [31]. Therefore, it is important to understand the molecular mechanisms of the response to heat and other abiotic stresses in *S*. *pohuashanensis*. Physiological and biochemical mechanisms of *S*. *pohuashanensis* under heat stress have been reported [32]. In recent years, transcriptomic and whole-genome sequencing have been used to analyze the molecular regulatory mechanisms involved in the response to heat stress in *S*. *pohuashanensis* [31,33,34]. Flavonoid biosynthesis and the Hsf–HSP pathway are important in the prevention of leaf sunburn in *S*. *pohuashanensis* [31,34]. Meanwhile, some studies have cloned and studied the functions of *SpHSP70s*, *SpHSP23.8*, and *SpHSP17.3* [35,36,37,38]. On this basis, SpHsfs family members were identified and analyzed, and it was found that most *SpHsfA* genes are upregulated under heat stress [33]. These advances further elucidated the adaptability of *S*. *pohuashanensis* to heat stress conditions at the molecular level and suggested a solution to the problem of leaf “sunburn” after *S*. *pohuashanensis* is introduced to low altitude areas. Because they are downstream genes of the Hsf-HSP pathway, the function of *HSP20s* is an important part of the molecular regulatory network of *S*. *pohuashanensis* in response to heat stress. However, analysis of the *HSP20* gene family in *S*. *pohuashanensis* has not yet been reported. Therefore, in this study, we identified the HSP20s family members of *S*. *pohuashanensis* and analyzed the expression of *SpHSP20s* in response to heat, drought, and salt stress to clarify their role in the regulation of abiotic stress and to provide a theoretical and information basis for the genetic improvement of *S. pohuashanensis.*

## 2. Materials and Methods

### 2.1. Plant Materials and Treatment

*S*. *pohuashanensis* was planted in the *Sorbus* germplasm resource nursery in Changping, Beijing. Two-year-old seedings of *S*. *pohuashanensis* with similar growth were selected and treated with heat (42 °C), drought, or NaCl salt stress. The same three families were used as experimental materials for each treatment and three plants from each family were used as replicates. Plants subjected to heat treatment were first pretreated with a temperature of 25 ± 2 °C, light intensity of 24,000 LX, and a 16/8 h light/dark cycle carried out in an artificial incubator (PGX-350D, SaiFu, Ningbo, China). Then, the temperature was gradually increased to 42 °C and was kept at that constant temperature until the end of the heat treatment (other conditions remained unchanged). Leaves were collected at 0, 2, 4, 6, 8, 12, 24, and 48 h after treatment, with 0 h being the control. Salt and drought treatments were conducted in a solar greenhouse in April 2021. The temperature in the greenhouse was no higher than 28 °C during the day, and no lower than 15 °C at night. Plants undergoing salt treatment were irrigated with 200 mM NaCl solution every 3 days; leaves were collected after 0, 1, 5, 10, and 15 d, with 0 d being the control. For plants subjected to drought treatment when the relative water content of the soil reached 80%, the water treatment was controlled and the leaves were collected after 0, 6, 9, 12, 15, and 18 d of drought treatment, with 0 d being the control. The leaf samples were obtained from the third to sixth leaves from the tops of three 2-year-old seedlings of *S*. *pohuashanensis* from three families. In addition, samples of the roots, stems, leaves, buds, flowers, and fruits of three 5-year-old grafted seedlings from the same family were collected for tissue-specific expression studies. All of the plant samples were quick-frozen in liquid nitrogen and stored at −80 °C for subsequent experiments.

### 2.2. Identification of HSP20 Genes in the S. pohuashanensis Genome

We downloaded the genome file from the genome database of the National Genomics Data Center (Accession No.: GWHBDNT00000000) and constructed a database of nonredundant proteins of *S*. *pohuashanensis.* Three methods were used to screen candidate genes: (1) with “HSP20” and “small heat shock protein” as key words, the candidate genes of *SpHSP20* were screened in the genome annotated document of *S*. *pohuashanensis*; (2) the HSP20 protein sequences of *A. thaliana* were downloaded from the TAIR (https://www.arabidopsis.org/index.jsp (accessed on 31 October 2021)) as a probe, with the candidate genes of *SpHSP20* obtained by searching the non-redundant protein database of *S*. *pohuashanensis* using blastP, and the threshold value was set to E-value < 1 × 10^−10^; and (3) the hidden Markov model (HMM) of the characteristic conserved domain of HSP20 (PF00011) was downloaded from Pfam (http://pfam.xfam.org/ (accessed on 16 March 2022)), and candidate genes of *SpHSP20* were obtained by searching the database of non-redundant proteins of *S*. *pohuashanensis* using the HMMER3 program based on HMM. To determine whether all candidate genes contained ACD, the protein sequences of the candidate genes of *SpHSP20* obtained using the three methods were submitted to the NCBI-CDD (http://www.ncbi.nlin.iiih.gov/Structure/cdd/wrpsb.cgi (accessed on 4 January 2022)), SMART (http://www.ebi.ac.uk/interpro/ (accessed on 4 January 2022)), and Pfam databases. In addition, the relative molecular weights of the genes were predicted. The candidate genes of *SpHSP20* without ACD and with a relative molecular weight outside 12–45 kDa were then removed. The same candidate genes that were screened using the three methods were used as the final members of the *HSP20* gene family of *S*. *pohuashanensis*.

### 2.3. Phylogenetic Analysis of SpHSP20 Genes

The sequences of HSP20s in *S*. *pohuashanensis*, *Populus trichocarpa, A. thaliana*, and *Malus domestica* (*Malus domestica*) were aligned using MEGA7.0, and they were then used to construct a phylogenetic tree with the neighbor-joining (NJ) method. The constructed phylogenetic tree was evaluated using the bootstrap method (repeated 1000 times). In addition, the Phylogenetic tree was visualized via the online tool iTOL (http://itol.embl.de/ (accessed on 11 April 2022)).

### 2.4. Analysis of Amino Acid Sequence, Gene Structure, and Conserved Motif of SpHSP20 Genes

The ExPasy online tools (http://www.expasy.org/ (accessed on 4 October 2021)) were used to predict the physicochemical properties of SpHSP20s. The online tool Protcomp (http://linux1.softberry.com (accessed on 22 December 2021)) was used to predict the subcellular localization of SpHSP20s. TBtools was used to show the structure of *SpHSP20s*. The MEME online tools (https://meme-suite.org/meme/tools/meme (accessed on 10 January 2022)) were used to analyze the conserved motif.

### 2.5. Chromosomal Location and Synteny Analyses

According to the annotation file of genomic structure information, the chromosomal position of *SpHSP20* genes was located using TBtools. Gene replication events were detected by MCScanX, and the synteny of *S*. *pohuashanensis* was visualized using TBtools.

### 2.6. Analysis of Cis-Elements in SpHSP20 Gene Promoters

We extracted 2000 bp upstream of the transcription initiation site ATG of *SpHSP20s* from the *S*. *pohuashanensis* genome database and submitted this data to the online database PlantCARE (http://bioinformatics.psb.ugent.be/webtools/plantcare/html/ (accessed on 16 April 2022)) for analyzing cis-acting elements in the *SpHSP20s’* promoter region.

### 2.7. Quantitative Real-Time PCR (qRT-PCR) and Expression Patterns Analysis

Total RNA was extracted using the EASYspin Plus Complex Plant RNA Kit (RA53, Aibosen Biology, Beijing, China), and it was detected by agarose gel electrophoresis. First-strand cDNA was synthesized according to the instructions of the *Evo M-MLV* RT Mix Kit with gDNA Clean for qRT-PCR (AG11728, Accurate Biology, Hunan, China). Finally, using the SYBR Green Premix Pro Taq HS qRT-PCR Kit (AG11701, Accurate Biology, Hunan, China) [33], qRT-PCR detection was performed on a CFX Connect^TM^ Real-Time PCR Detection System (PXF2080, Bio-Rad, CA, USA). *Spβ-actin* was used as an internal reference gene, and the primer sequences are listed in Appendix A. The quantitative results were analyzed using the 2^−ΔΔCT^ method [39].

## 3. Results

### 3.1. Identification of HSP20 Genes in S. pohuashanensis

A total of 57 *SpHSP20* candidate genes were screened through a keyword search, HMM retrieval, and blastP alignment. After conservative domain prediction and molecular weight screening, 19 candidate genes were removed after they were determined to not belong to the *HSP20* gene family. Finally, 38 *SpHSP20* gene family members were obtained. All *SpHSP20* genes had an HSP20 domain (Appendix A). The sequence of the *SpHSP20s’* CDs, protein, and promoter are listed in Appendix A. The positions on the chromosomes of SpHSP20s are listed in Table 1. The predicted physical and chemical properties show that the number of amino acids (AAs) encoded by members of the *SpHSP20* gene family ranged from 111 (SpHSP12.7) to 285 (SpHSP30.6), and the molecular weight ranged from 12.68 kDa (SpHSP12.7) to 31.54 kDa (SpHSP31.5). Naming was based on the molecular weights. The isoelectric point (pI) was between 4.71 (SpHSP16.7B) and 9.56 (SpHSP17.0), and most of the proteins were acidic. The instability index (in which a value greater than 40 is considered to indicate an unstable protein) indicated that there are 31 unstable proteins in the *SpHSP20* gene family, while the remaining seven SpHSP20s are stable. To understand SpHSP20s’ protein function, we used a website to predict the subcellular localization of SpHSP20s. The results indicate that SpHSP20s may play a role in these localizations.

### 3.2. Phylogenetic Analysis of SpHSP20 Genes

To explore the phylogenetic relationship of the *HSP20* gene family among different species, HSP20 family members have been classified in three other species (19 HSP20s from *Arabidopsis*, 41 HSP20s from *Malus domestica*, and 30 HSP20s from *P. trichocarpa*), and 38 HSP20s from *S. pohuashanensis* were arranged in a phylogenetic tree (Figure 1). With the exception of MdHSP20-4, MdHSP20-11, and SpHSP20.3A, the other HSP20s were clustered into 12 subfamilies (40 CIs, 19 CIIs, 5 CIIIs, 4 CIVs, 8 CVs, 2 CVIs, 1 CVII, 5 MIs, 11 MIIs, 10 Ps, 6 Pos, and 10 ERs). There was obvious clustering of HSP20s in *S. pohuashanensis*, *Arabidopsis*, *P. trichocarpa*, and *Malus domestica*. From an evolutionary perspective, the genetic relationship between *S. pohuashanensis* and *Malus domestica* is the closest, followed by *P. trichocarpa* and *Arabidopsis*. SpHSP20.3A and two unclassified MdHSP20s separately clustered into the same branch and were unclassified, and the other 37 SpHSP20s were divided into 10 subfamilies. Nineteen (50%) SpHSP20s were classified as CI–CV, but none of them were clustered to CVI or CVII of the 12 subfamilies.

### 3.3. Conserved Motifs and Gene Structure Analysis of SpHSP20 Genes

As shown in Figure 2a, except for SpHSP20.3A, 37 SpHSP20s were divided into 10 subgroups based on the homology of amino acid sequences. A total of 10 conserved motifs were screened out using MEME online tools (Figure 2b). The logos and amino acid sequences of the conserved motifs are listed in Appendix A. Except for SpHSP20.3B, which contains only one motif, SpHSP20s contain 3–7 motifs. Almost all HSP20s of the C and ER subclasses contain motifs 1, 2, 3, 4, 5, and 8. In addition, SpHSP20s in the same subgroup have similar motif compositions, whereas SpHSP20s in different subgroups have different motif compositions. The farther apart the subgroups, the greater the difference in motif composition. These results indicate that *SpHSP20* genes have undergone significant differentiation through evolution.

The structures of the 38 *SpHSP20* genes were visualized using TBtools (Figure 2c). Fifteen (39.5%) *SpHSP20* genes had no introns, 21 (55.3%) had one intron, and 2 (5.2%) had two introns; 21 (55.3%) *SpHSP20* genes had no untranslated regions (UTRs), 12 (31.6%) had two UTRs, 4 (13.1%) had three UTRs, and 1 (SpHSP19.8) had one UTR. Furthermore, 94.8% and 86.9% of the SpHSP20s did not contain or contained a few introns and UTRs, respectively, indicating that the structure of the *SpHSP20* gene family is relatively simple.

### 3.4. Chromosomal Location, Gene Duplication, and Synteny Analyses

Based on the annotation file of the *S. pohuashanensis* genome, the chromosomal positions of *SpHSP20* genes were located using TBtools (Figure 3). It was found that 38 *SpHSP20* genes were unevenly distributed on 16 chromosomes of *S. pohuashanensis* (except Chr02), among which Chr01 and Chr08 contain the most *SpHSP20* genes (five); Chr07 and Chr17 each have four *SpHSP20s*; Chr09, Chr10, and Chr15 each have three *SpHSP20s*; Chr02 and Chr04 have two *SpHSP20s*; the remaining seven *SpHSP20s* are distributed on the other seven chromosomes. Moreover, we detected gene duplication events using MCScanX and obtained genome-wide tandem and segmental duplicated gene pairs (Appendix A). The TBtools software was used to visualize the syntenic analysis results of the *SpHSP20* gene family. The results show that six pairs of *SpHSP20* genes were regarded as tandemly duplicated genes located on Chr01, Chr07, Chr08, and Chr15 (Figure 3, Appendix A). In addition, except for 13 genes (*SpHSP16.0B*, *SpHSP16.4*, *SpHSP17.0*, *SpHSP17.3C*, *SpHSP17.8*, *SpHSP18.0*, *SpHSP20.3A*, *SpHSP24.4*, *SpHSP24.7*, *SpHSP26.0*, *SpHSP27.0*, *SpHSP30.6*, and *SpHSP31*.0), the *SpHSP20* genes had 28 segmental duplication gene pairs (Figure 4, Appendix A). These results indicated that tandem and segmental duplications promoted the formation of the *SpHSP20* gene family to different degrees.

We also compared the syntenic blocks of the genomes of *S. pohuashanensis*, *Arabidopsis*, and *Malus domestica*, and found that many *HSP20* genes are homologous (Figure 5). There were more syntenic gene pairs of the *HSP20* gene family between *S. pohuashanensis* and *Malus domestica* (the syntenic gene pairs are shown in Appendix A). This indicates that *S. pohuashanensis* and *Malus domestica* are more closely related, and this is consistent with phylogenetic analysis.

### 3.5. Analysis of Cis-Elements in SpHSP20 Gene Promoters

To explore the potential physiological functions of SpHSP20s, a promoter region was selected at a location 2000 bp upstream of each *SpHSP20* transcription start site (the codon ATG), and this was uploaded to the PlantCARE online database to analyze cis-acting elements. The promoters of *SpHSP20* genes contain twenty kinds of cis-acting elements (Figure 6a), of which five are related to growth and development, six are related to hormone responsiveness, two are related to substance metabolism and synthesis, and seven are related to the stress response. All predicted raw data are listed in Appendix A, including other cis-acting elements that were not analyzed. Among the stress-related cis-acting elements, 33 contained anaerobic induction cis-acting elements, 17 contained drought inducibility elements, 13 contained HSE, and 23 contained low temperature-responsive elements. These results suggest that SpHSP20s have a potential stress response under abiotic stresses such as drought and extreme temperature. Abscisic acid, salicylic acid, methyl jasmonate, auxin, and gibberellin-related cis-acting elements were identified among the hormone-related cis-acting elements. Some *SpHSP20* promoters contain cis-acting elements involved in growth and development, such as circadian control and meristem expression. All *SpHSP20* promoters contained light-responsive elements. A few *SpHSP20* promoters contained cis-acting elements related to the regulation of flavonoid biosynthesis genes and zein metabolism (Figure 6b). These results show that the *SpHSP20* gene family is involved in plant growth and development, as well as the response to various abiotic stresses, and that the gene family has complex biological functions.

### 3.6. Expression Patterns of SpHSP20 Genes under Heat, Salt, and Drought Stress

We downloaded previous transcriptome data (Bioproject ID: PRJNA716887) from the NCBI SRA (Sequence Read Archive) database to evaluate the expression pattern of *SpHSP20s* under natural high temperature stress using FPKM (fragments per kilobase of transcript per million fragments mapped) values. Compared with the control, almost all *SpHSP20s* were upregulated after high temperature stress, and the expression level of most *SpHSP20s* reached the highest within a short time, while some *SpHSP20s* showed multiple expression peaks during the stress process (Appendix A). In addition, the sample environmental data for transcriptome sequencing were provided in Appendix A. A heatmap of 38 *SpHSP20* genes was constructed using relative expression values to evaluate their expression levels under heat stress (Figure 7, Appendix A). Compared with those at 0 h of treatment, after heat stress treatment at 42 °C, all *SpHSP20s* were upregulated to different degrees. The heatmap shows that the 38 *SpHSP20* genes clustered into five groups, with most *SpHSP20s* (79%) belonging to clusters B and C. The expression levels of clusters A, B, and C all peaked at 2 h, and then the expression levels of cluster A (*SpHSP21.5* and *SpHSP27.2*) began to decline, whereas the expression levels of clusters B and C decreased at 8–12 h. The expression levels of clusters A, B, and C increased again after 24 h and the expression levels of cluster B at 24 h were similar to those at 2 h. The expression of cluster D reached its peak at 6–8 h, and then gradually decreased. Interestingly, the expression of cluster E (*SpHSP16.7A*, *SpHSP16.7B*, *SpHSP17.3A*, and *SpHSP23.5*) was downregulated at 2 h, and then it began to increase, gradually reaching its peak after 8 h. Meanwhile, the expression levels of *SpHSP24.4*, *SpHSP24.7*, and *SpHSP27.0* in cluster B were consistently high during the entire heat stress process, suggesting that these three genes may be important in the response of *S. pohuashanensis* to heat stress. The experimental results were basically consistent with the transcriptome data.

The heatmap shows the expression patterns of 38 *SpHSP20* genes under salt stress clustered into four groups (Figure 8, Appendix A). Clusters A, B, C, and D contained 3, 19, 5, and 11 *SpHSP20s*, respectively. The expression of clusters A and B reached a peak at 15 d, and, compared with cluster B, the expression of cluster A also increased significantly at 5 d. The expression of cluster C tended to increase first and then decrease, reaching a peak at 5 d. The expression level of cluster D showed a trend of increased → decreased → increased; it reached the peak at 5 d, decreased significantly at 10 d, and then was upregulated at 15 d. In conclusion, the *SpHSP20* gene family showed a diverse expression trend during different periods of salt stress.

Similarly, the heatmap shows the expression patterns of 38 *SpHSP20* genes under drought stress clustered into three groups (Figure 9, Appendix A). The expression of cluster A peaked at 9 d, after which it gradually decreased, and then increased again at 18 d. The expression patterns of most *SpHSP20* gene family members under drought stress belonged to clusters B and C. The expression of clusters B and C reached a peak at 18 d, but compared with cluster B, the expression of cluster C was significantly increased at 9 d. In general, all *SpHSP20s* were upregulated at 18 d of drought stress treatment, and some *SpHSP20s* (such as clusters A and C) were also significantly upregulated at 9 d.

### 3.7. Tissue-Specific Expression of SpHSP20 Genes

Tissue-specific expression analysis revealed significant differences in the constitutively specific expression of *SpHSP20* gene family members in different organs (Figure 10, Appendix A). The heatmap shows the expression patterns of 38 *SpHSP20* genes in different organs clustered into six groups. All *SpHSP20* genes were expressed in at least one organ. Except for cluster A (*SpHSP16.0A*), the expression level of *SpHSP20s* in buds was the lowest, and this was only slightly higher in flowers. However, most *SpHSP20s* (including clusters B, C, D, E, and F) showed the highest expression in fruits. In addition, clusters B, D, and E were highly expressed in roots and stems, and clusters B, C, and F were also expressed in leaves. These differences may be related to the involvement of SpHSP20s in different growth and development processes.

## 4. Discussion

HSP20s are considered the most abundant family of HSPs involved in plant responses to stress [12]. Many studies have shown that HSP20s could avoid irreversible aggregation of denatured proteins and participate in the establishment of plant tolerance to heat and other abiotic stresses [16]. With the development of whole-genome sequencing technology, an increasing number of *HSP20* gene families have been identified; for example, 51, 39, 42, 19, 35, 36, 41, 44, and 48 *HSP20* gene family members have been identified in soybean [21], rice [15], tomato [20], *Arabidopsis* [23], pepper [22], *P. trichocarpa* [40], *Malus domestica* [41], watermelon (*Citrullus lanatus*) [42], and grape (*Vitis vinifera*) [43]. These results are conducive to the identification of HSP20 family members in *S. pohuashanensis*.

In this study, we identified 38 HSP20s in *S. pohuashanensis*. Analysis of the physical and chemical properties showed that the 38 SpHSP20 proteins had significant differences in molecular weight and theoretical electric points. The prediction of subcellular localization provides a reference for the study of protein function. We used the subcellular localization prediction website to predict the possible subcellular localization of SpHSP20s. The results showed that 38 SpHSP20s were localized in various locations such as the cytoplasm, nucleus, chloroplast, endoplasmic reticulum, peroxisome, and mitochondria (Table 1). These results indicate that the sequence characteristics of the *SpHSP20* gene family members gradually differentiated during evolution. These differences may have been caused by gene duplication in the *SpHSP20* gene family. Gene family evolution involves recombination (including gene conversion), gene duplication, and gene loss [12]. Gene duplication is a major driving force for the evolution of plant gene families [44]. Tandem and segmental duplications often affect gene family size and distribution [45]. In this study, 38 *SpHSP20s* were unevenly distributed on 16 chromosomes, and most *SpHSP20s* were located at the ends of chromosomes, which might be related to duplication events in the *SpHSP20* gene family. Synteny analysis reveals that there were six tandem duplicated gene pairs in 38 *SpHSP20s* (Figure 3), and 25 (65.8%) *SpHSP20s* were segmentally duplicated genes (Figure 4), indicating that segmental duplication more significantly impacted the evolution of the *SpHSP20* gene family. This is similar to the results of a previous *Malus domestica HSP20* gene family analysis [41]. In addition, syntenic analysis of the *HSP20* gene family in *S. pohuashanensis, Arabidopsis*, and *Malus domestica* shows that the *SpHSP20* gene family had more syntenic genes with *Malus domestica*, which further indicates that the genetic relationship between *S. pohuashanensis* and *Malus domestica* is closer.

To explore the evolutionary relationship of the *HSP20* gene family among different species, a phylogenetic tree containing 128 HSP20 protein sequences from the model plant *Arabidopsis*, the woody plant *P. trichocarpa*, *S. pohuashanensis*, and *Malus domestica* belonging to the same subfamily with *S. pohuashanensis* was constructed (Figure 1). Following the classification of HSP20s in three species, the other 38 SpHSP20s were clustered into 10 subfamilies except for SpHSP20.3A, and 19 (50%) SpHSP20s were classified as CI–CV which indicates that the cytoplasm may be the main functional region of SpHSP20s [46]. These results are similar to those of the *Malus domestica HSP20* gene family cluster [41]. However, compared with the *HSP20* gene family of *P. trichocarpa* [40], it lacks CVI; it also lacks CVI and CVII compared with the *HSP20* gene family of *Arabidopsis* [23]. This may have resulted from gene loss events during the evolution of gene families. Similarly, the *HSP20* gene family in pepper lacks CIV and CV [22], while CIV and CVII are absent from the *HSP20* gene family of rice [13]. Therefore, gene expansion and loss are common among plant species.

Gene structure (exon-intron) plays an important role in multi-gene family evolution, and genes with few or no introns can be activated quickly under different stresses [47]. Studies have shown that alternative splicing plays a role in plants’ response to abiotic stress [48]. Analysis of the gene structure (Figure 2c) shows that 94.8% of *SpHSP20s* contain only one intron (55.3%) or no introns (39.5%), indicating that the splicing arrangement of exon–intron of *SpHSP20* genes was conservative. This is similar to previous studies on grape [43], pepper [22], and tomato [20]. In this study, most *SpHSP20s* of CII and ER did not have introns, whereas those of MI and MII all had introns, similar to the results in grape [43], rice [15], and soybean [21]. However, the gene structure of *SpHSP20* genes in CI and CIII was different from that in grape [43], suggesting that the structure of *HSP20* genes is distinct from different species. Moreover, most SpHSP20s (81.6%) had a stability index greater than or close to 40 (Table 1), indicating that those SpHSP20s are unstable. Instability is considered a common characteristic of stress response proteins [21,22], and it may be related to the finding that *SpHSP20* genes can be rapidly induced [49]. For further study on the evolution of the *HSP20* gene family, we searched for conserved motifs in SpHSP20s and found that most SpHSP20s contained 3–7 conserved motifs and almost all SpHSP20s had motif 1 and motif 2. We also found that SpHSP20s of the same subfamily have similar motif compositions (Figure 2b), indicating that they are closer in terms of evolutionary relationships. This is similar to *Malus domestica* [41] and tomato [20]. In addition, the motif compositions of the P, Po, and cytoplasmic subfamilies SpHSP20s were significantly different, suggesting that SpHSP20s were differentiated in the evolutionary process and this led to functional differences.

Cis-acting elements in gene promoters are essential for plant physiological responses and environmental stresses [50]. We analyzed the cis-acting elements in the *SpHSP20s’* promoter region and found that there were numerous elements related to plant growth and development, stress response, and hormones (Figure 6). Among the stress-related homeopathic elements, 13 *SpHSP20* promoters contained HSE; and 23 contained low-temperature response elements, suggesting that SpHSP20s have a potential stress response under temperature stress and that SpHSP20s might be more important in the response to low-temperature stress than in heat stress. In addition, 33 *SpHSP20* promoters contained cis-acting elements of anaerobic induction, and 17 *SpHSP20* promoters contained drought-inducibility elements, which may be connected with long-term adaptation to low temperature, hypoxia, and drought environments of *S. pohuashanensis* growing at high altitudes. There are also many cis-acting elements related to hormones in the promoter region of *SpHSP20s*, suggesting that the *HSP20* gene family has a complex regulatory relationship with the stress response and hormone signal transduction. However, the regulatory network of SpHSP20s in hormone signal transduction and stress responses remains unclear. Moreover, various cis-acting elements involved in growth and development were present in the *SpHSP20s’* promoter region, and it was evident that all *SpHSP20* promoter regions had different numbers of light-responsive cis-acting elements, further suggesting that the *HSP20* gene family is a key factor in plant growth and development. These results are consistent with those for *Malus domestica* [41], pepper [22], and grape [43].

Studies have shown that HSP20s are involved in plant responses to various abiotic stressors [22,42,43,51]. In this study, we analyzed the expression patterns of *SpHSP20s* under heat (42 °C), NaCl salt treatment, and drought stresses, and we found that almost all *SpHSP20s* were induced by high temperature. Previous studies have shown that genes with few or no introns were expressed at higher levels [52,53]. To respond promptly to various environmental stresses, genes must be activated rapidly, and this is achieved by gene structures with fewer introns [47]. In this study, the expression levels of most *SpHSP20s* reached their peak at 2 h under heat stress (Figure 7). Similarly, most *HSP20s* in pepper [22], potato (*Solanum tuberosum*) [51], and *Malus domestica* [41] are upregulated under short-term heat stress. Interestingly, we found that three genes, *SpHSP24.4*, *SpHSP24.7*, and *SpHSP27.0*, were always highly expressed during the entire heat stress process, suggesting the importance of these three genes in the response to heat stress. Both HSPs and Hsfs are involved in the heat stress reaction network, and HSP expression is activated by the binding of Hsfs to HSE elements in the *HSP* promoter region during heat shock. The number of Hsfs varies widely among species; there are only 1 to 4 *Hsfs* genes in animals, whereas *Arabidopsis* has 21 *Hsfs* genes [54]. A study showed that 33 *SpHsf* genes are present in *S. pohuashanensis*, and most *SpHsfA* genes are upregulated under heat stress [33]. This leads to the inference that there is a complex regulatory mechanism between SpHsfs and SpHSP20s. However, the cis-acting elements in the *SpHSP20* promoter regions were analyzed, and it was found that not all *SpHSP20s* had HSE, and more *SpHSP20s* had low temperature-responsive elements. Similarly, in *Malus domestica* [41], only part of the *HSP20* promoter regions had HSE. We speculate that this might result from the loss of HSE elements in the evolution of *S. pohuashanensis* growing in high altitude areas and being exposed to prolonged low temperatures, causing the loss of heat resistance of *S. pohuashanensis*. Meanwhile, some *SpHSP20s* without HSE were also highly expressed under heat stress, indicating that these *SpHSP20s* may not be directly activated by SpHsfs, and that there may be interactions between different SpHSP20s to jointly respond to heat stress. However, in *Cucurbita moschata*, the expression of *HSP20* in roots did not significantly change under heat stress. After 3 h of heat stress, the expression of most of the *CmoHSP20* in the leaves was significantly upregulated, but there was also downregulated *HSP20* expression and no significant change in *HSP20* expression during heat stress [55]. These results indicate that the expression patterns of HSP20 in response to heat stress were different in different species and plant organs.

The expression patterns of *SpHSP20s* were similar under salt and drought stress, and most *SpHSP20s* were upregulated under salt (Figure 8) and drought stress (Figure 9). This is similar to the expression pattern of *HSP20s* in *Salix suchowensis* and switchgrass (*Panicum virgatum*) under salt stress [56,57]. Nevertheless, in *Cynodon transvaalensis*, only several *HSP20s* were expressed under drought stress and salt stress [58]. In addition, in *Pyropia yezoensis*, the expression of almost all *HSP20s* was downregulated under dehydrated stress and upregulated after rehydration [59]. In addition, multiple *SpHSP20s* exhibit the same expression pattern under the same abiotic stress, and many *SpHSP20s* can be induced to be expressed by multiple abiotic stresses, suggesting that SpHSP20s may cross-act and interact in response to different abiotic stresses and that they have a complex regulatory network in response to abiotic stresses.

HSP20s participate in heat, cold, drought, salt, and other environmental stresses and also participate in embryogenesis, seed germination, fruit ripening, and other growth and development processes [60,61]. In the tissue-specific studies of the *HSP20* gene family in potato, *Salix suchowensis,* and grape, it was found that the expression level of *HSP20s* in the reproductive organs (such as flowers and fruits) was higher than that in vegetative organs (such as leaves and stems), and the expression levels of some *HSP20s* were higher in grapefruits during the color turning stage [43,51,56]. In this study, the tissue-specific expression of *SpHSP20s* showed that almost all *SpHSP20s* were expressed at the lowest level in buds and at the highest level in fruits. Only a few *SpHSP20s* had low expression levels in flowers, and most *SpHSP20s* had intermediate expression levels in roots, stems, and leaves (Figure 10). They showed obvious specificity. The expression of *HSP20s* in roots, stems, leaves, and other organs of *S. pohuashanensis* indicates that SpHSP20s are involved in the growth and development of plants, and the expression level of *SpHSP20s* is the highest in fruits. Studies have shown that the *COP1LIKE* gene in tomato can delay fruit discoloration by reducing ethylene production [62]. So, we speculate that SpHSP20s may be related to plant hormone regulation. In addition, we speculate that SpHSP20s may also participate in the synthesis of nutrients and medicinal ingredients in the fruit of *S. pohuashanensis* [63,64]; however, there is no research in this area, and this issue must be further investigated.

This study on the expression pattern of the *SpHSP20* gene family under different abiotic stresses provides the gene selection for using molecular breeding methods to solve the problem of the leaf “sunburn” phenomenon and the poor growth of *S. pohuashanensis* introduced to low altitude areas. For example, the expression levels of *SpHSP24.4*, *SpHSP24.7*, and *SpHSP27.0* were consistently high during the entire heat stress process, suggesting that these three genes may be important in the response of *S. pohuashanensis* to heat stress. The analysis of cis-acting elements in promoter regions provides a theoretical basis for further analysis of the regulatory mechanisms of other factors on SpHSP20s; anaerobic induction cis-acting elements, drought inducibility elements, low temperature-responsive elements, as well as other stress-related cis-acting elements, may be transcription factor binding sites. Many hormone-related cis-acting elements suggest that SpHSP20s may be closely related to hormone regulation or signal transduction. However, these speculations need to be studied further.

## 5. Conclusions

In our study, we identified 38 *SpHSP20* genes according to the genome data of *S. pohuashanensis*. These 38 *SpHSP20* genes were unevenly located on 16 chromosomes. According to phylogenetic analysis and subcellular localization prediction, they were divided into 10 subfamilies. To better explore the evolutionary relationships between the *HSP20* gene family members, we analyzed the structure, conserved motif, cis-acting elements, and synteny. In addition, we studied the expression patterns of *SpHSP20s* in different organs and under different stress treatments, providing information for further study on the function of *HSP20* genes. These results provide a theoretical basis for analyzing the molecular regulatory mechanisms under heat stress and provide a theoretical foundation for the introduction of *S. pohuashanensis* to low altitude areas and its genetic improvement.

## Figures and Tables

**Figure 1 genes-13-02241-f001:**
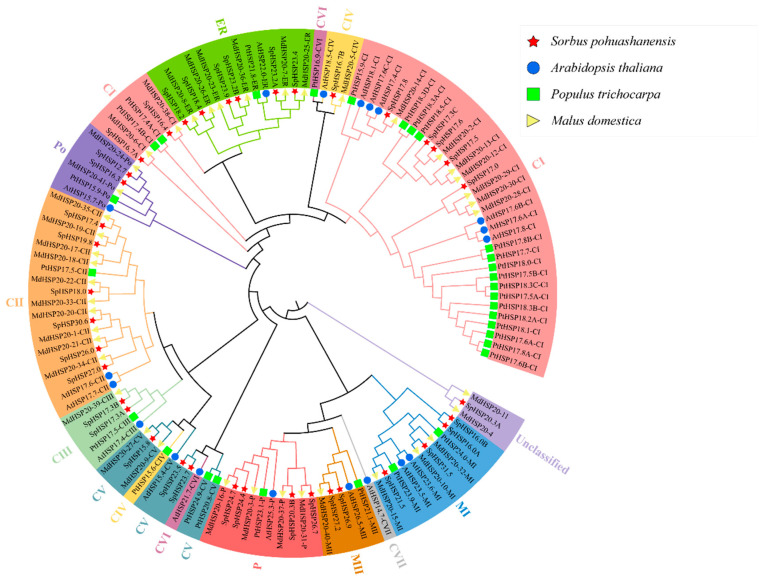
Phylogenetic tree of HSP20 proteins from *Sorbus pohuashanensis*, *Arabidopsis thaliana*, apple, and *Populus trichocarpa.* The 12 subfamilies were distinguished with different colors. The four species are distinguished by different colored shapes. The phylogenetic tree was constructed using MEGA7.0 based on the NJ method and visualized via the online tool iTOL; bootstrap was 10,000 replicates.

**Figure 2 genes-13-02241-f002:**
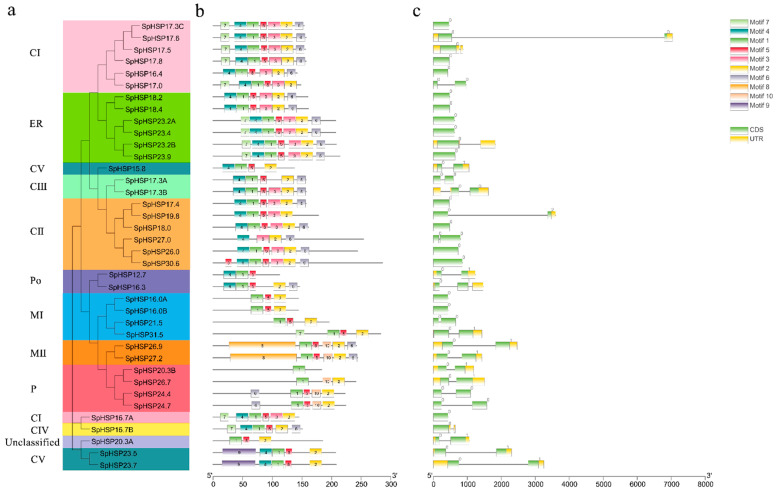
Phylogenetic tree, conserved motifs and structure analyses of *SpHSP20s.* (**a**) Phylogenetic tree of *SpHSP20s.* (**b**) Conserved motifs analyses of SpHSP20 proteins. Different colored boxes represented the different types of motifs. The amino acid sequences of the motifs 1–10 are listed in Appendix A (**c**) Gene structure of *SpHSP20s*. CDS sequences are represented by yellow round-corner rectangles and introns by grey lines; UTRs are shown with green boxes.

**Figure 3 genes-13-02241-f003:**
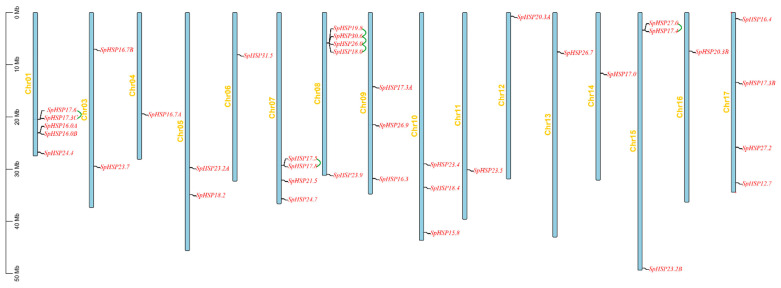
Chromosomal locations and tandem duplications of *SpHSP20* genes on grape chromosomes. Black lines indicated the gene position. The six green arcs represent six tandem duplicated gene pairs which are listed in Appendix A.

**Figure 4 genes-13-02241-f004:**
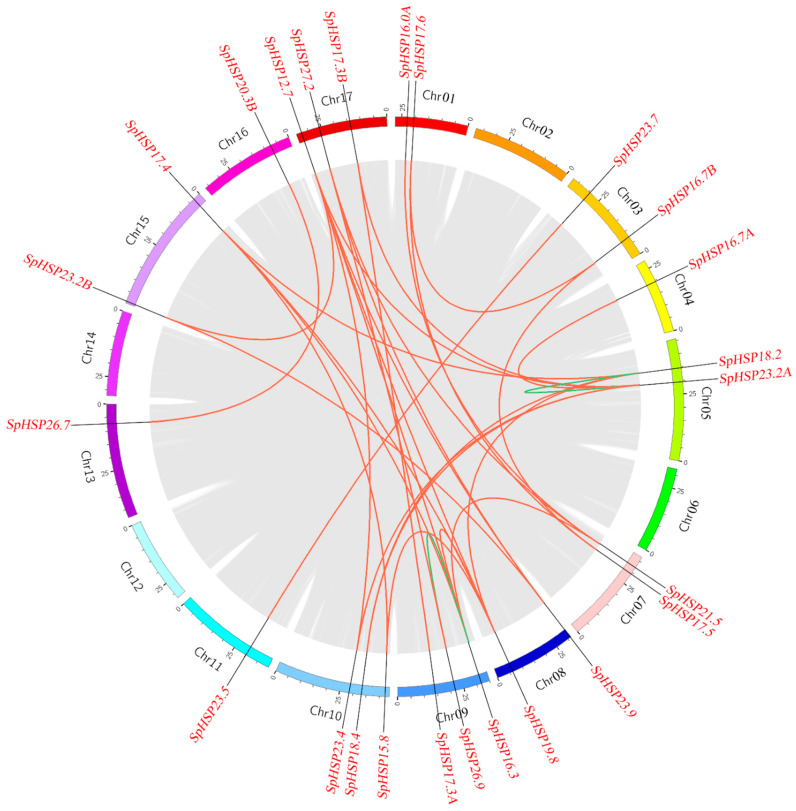
Syntenic relationships of the *SpHSP20* genes. Grey lines indicate all homologous blocks in the *Sorbus pohuashanensis* genome, and the red and green lines indicate segmental duplicated *SpHSP20* gene pairs on the same and different chromosomes. The segmental duplicated gene pairs are listed in Appendix A.

**Figure 5 genes-13-02241-f005:**
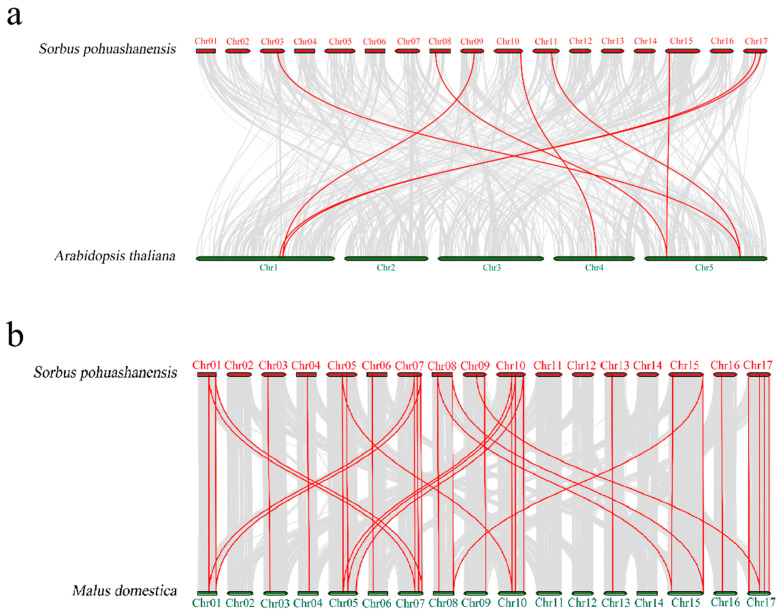
Syntenic analysis of *HSP20* gene in *Sorbus pohuashanensis*, apple, and *Arabidopsis thaliana.* (**a**) Syntenic analysis of *HSP20* gene between *Sorbus pohuashanensis* and *Arabidopsis thaliana*. (**b**) Syntenic analysis of *HSP20* gene between *Sorbus pohuashanensis* and apple. Grey lines indicate the syntenic blocks within *Sorbus pohuashanensis* and other plant genomes, and the red line indicates syntenic gene pairs. The syntenic gene pairs are listed in Appendix A.

**Figure 6 genes-13-02241-f006:**
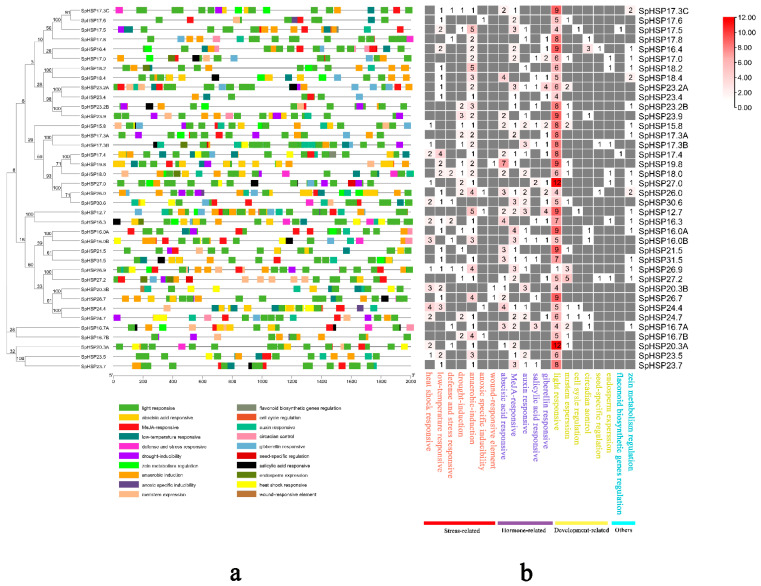
Cis-element analysis of *SpHSP20* gene promoters. (**a**) The different colored blocks represent the different types of cis-acting elements and their locations in each *SpHSP20* gene. (**b**) The number of each cis-acting element in the promoter region of each *SpHSP20* gene.

**Figure 7 genes-13-02241-f007:**
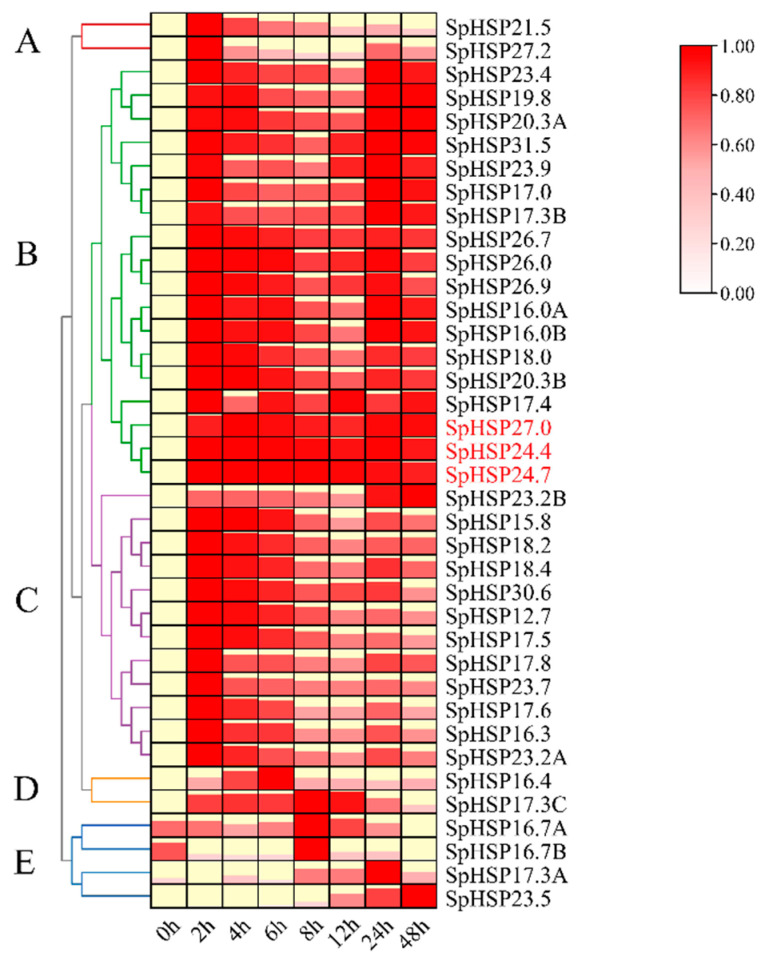
The expression patterns of *SpHSP20s* under heat stress (42 °C) at different times. The graphic size and color represent the level of expression of each gene in each sample. Different patterns of expression are represented by branches of different colors and are numbered A–E. *Spβ-actin* is used as an internal control. qRT-PCR data are shown relative to 0 h. The relative expression levels were calculated using the 2^−ΔΔCT^ method and normalized using log2.

**Figure 8 genes-13-02241-f008:**
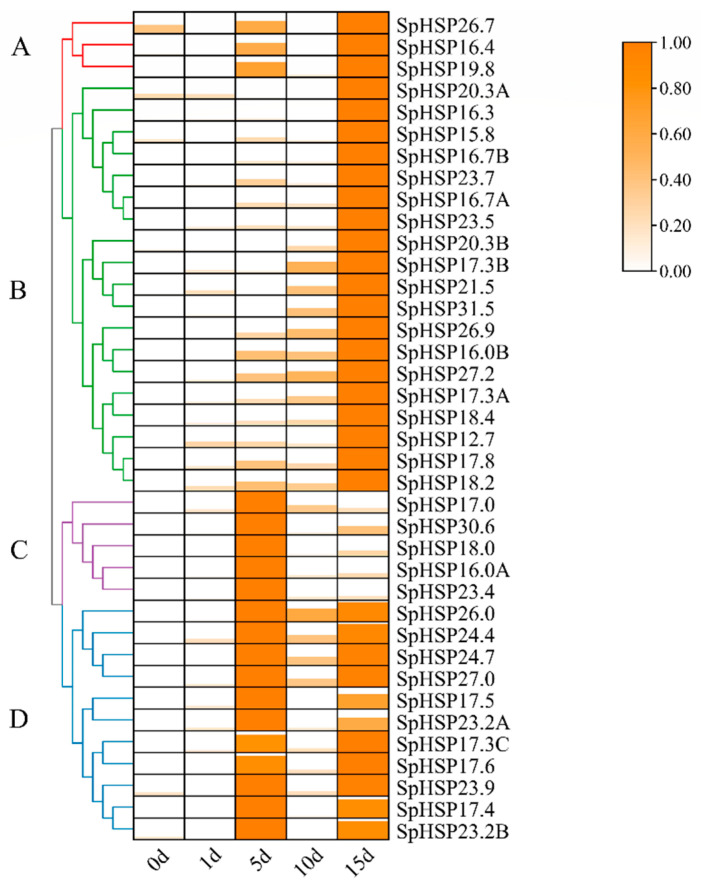
The expression patterns of *SpHSP20s* under Nacl salt stress (200 mM) at different times. The graphic size and color represent the level of expression of each gene in each sample. Different patterns of expression are represented by branches of different colors and are numbered A–D. *Spβ-actin* is used as an internal control. qRT-PCR data are shown relative to 0 d. The relative expression levels were calculated using the 2^−ΔΔCT^ method and normalized using log2.

**Figure 9 genes-13-02241-f009:**
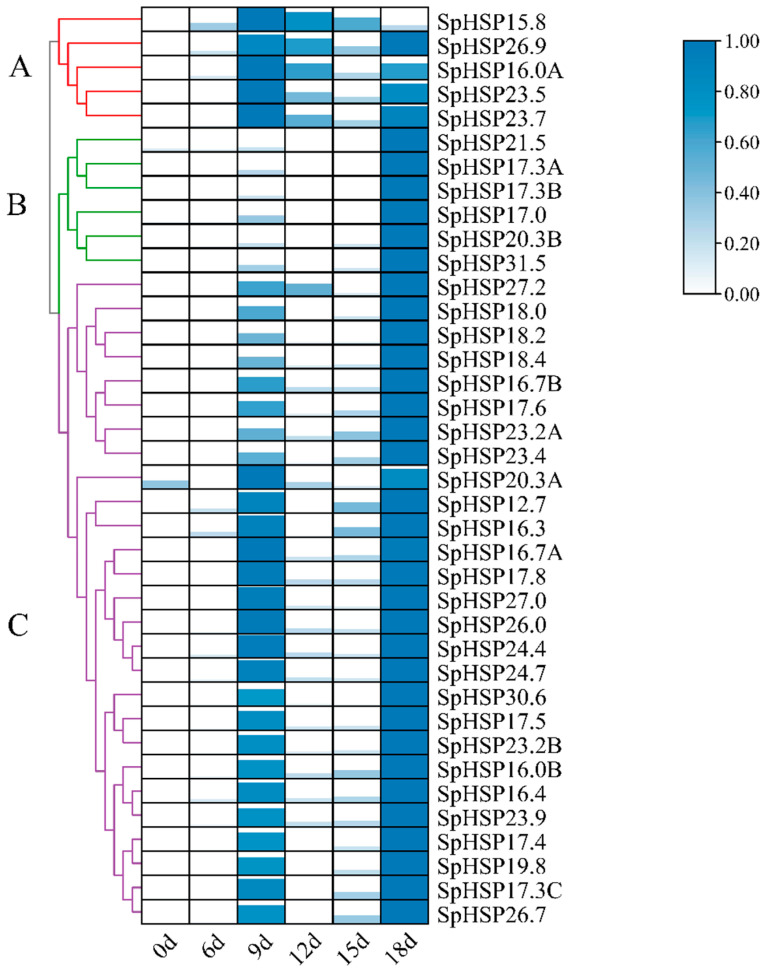
The expression patterns of *SpHSP20s* under drought stress at different times. The graphic size and color represent the level of expression of each gene in each sample. Different patterns of expression are represented by branches of different colors and are numbered A–C. *Spβ-actin* is used as an internal control. qRT-PCR data are shown relative to 0 d. The relative expression levels were calculated using the 2^−ΔΔCT^ method and normalized using log2.

**Figure 10 genes-13-02241-f010:**
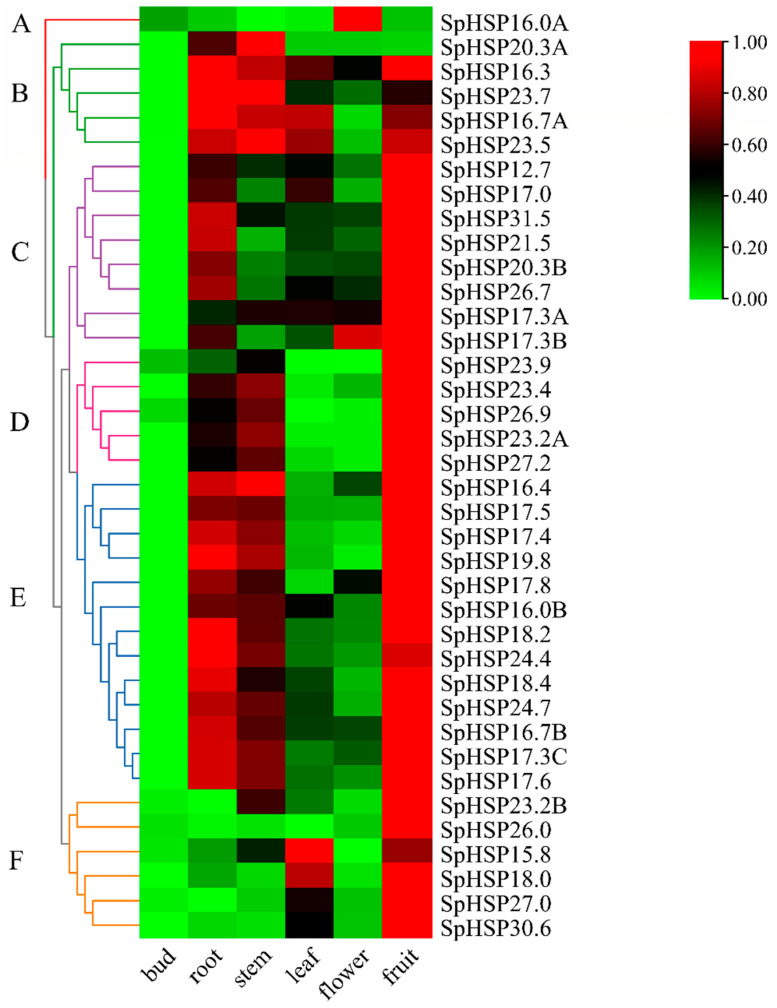
The expression patterns of *SpHSP20s* in different organs. The different colors represent the level of expression of each gene in each sample. Different patterns of expression are represented by branches of different colors and are numbered A–F. *Spβ-actin* is used as an internal control. qRT-PCR data are shown relative to the buds. The relative expression levels were calculated using the 2^−ΔΔCT^ method and normalized using log2.

**Table 1 genes-13-02241-t001:** Information on members of the HSP20 gene family of *Sorbus pohuashanensis*.

Gene Name	Gene ID	Chr	Chromosome Position	AA	MW (KDa)	pI	Stability Index	Subcellular Localization
*SpHSP12.7*	Sp17G024510.1	17	32,629,484–32,630,718	111	12.67843	5.93	54.41	Peroxisomal
*SpHSP15.8*	Sp10G029800.1	10	42,110,310–42,111,362	136	15.7539	5.37	73.02	Cytoplasmic
*SpHSP16.0A*	Sp01G015920.1	01	23,033,251–23,033,682	143	16.00126	6.1	30.47	Mitochondrial
*SpHSP16.0B*	Sp01G015940.1	01	23,051,500–23,051,931	143	16.03922	5.6	37.15	Mitochondrial
*SpHSP16.3*	Sp09G023920.1	09	31,776,066–31,777,923	145	16.31363	8.02	50.1	Peroxisomal
*SpHSP16.4*	Sp17G001570.1	17	1,170,468–1,170,893	141	16.39386	7.82	34.52	Cytoplasmic
*SpHSP16.7A*	Sp04G012430.1	04	19,406,854–19,407,288	144	16.69707	6.71	50.81	Nuclear
*SpHSP16.7B*	Sp03G006770.1	03	7,035,379–7,036,024	150	16.72572	4.71	57.84	Nuclear
*SpHSP17.0*	Sp14G008870.1	14	11,649,378–11,650,338	147	17.03848	9.56	51.98	Nuclear
*SpHSP17.3A*	Sp09G015010.1	09	14,210,375–14,210,971	156	17.28986	7.72	35.56	Nuclear
*SpHSP17.3B*	Sp17G013620.1	17	13,391,460–13,393,080	156	17.33689	7.73	41.83	Cytoplasmic
*SpHSP17.3C*	Sp01G012780.1	01	20,465,985–20,466,446	153	17.34467	6.18	47.22	Nuclear
*SpHSP17.4*	Sp15G004370.1	15	3,391,645–3,392,115	156	17.43811	5.58	39.53	Cytoplasmic
*SpHSP17.5*	Sp07G019210.1	07	29,273,801–29,274,672	155	17.52473	5.56	52.97	Cytoplasmic
*SpHSP17.6*	Sp01G012770.1	01	20,440,281–20,450,225	156	17.62492	5.84	51.77	Nuclear
*SpHSP17.8*	Sp07G019220.1	07	29,289,942–29,290,412	156	17.81794	6.39	67.34	Nuclear
*SpHSP18.0*	Sp08G006240.1	08	5,860,244–5,860,726	160	18.00361	5.97	45.72	Cytoplasmic
*SpHSP18.2*	Sp05G022540.1	05	34,877,634–34,878,113	159	18.22156	5.72	45.73	Chloroplast
*SpHSP18.4*	Sp10G020960.1	10	33,482,466–33,482,948	160	18.42399	6.32	45.11	Cytoplasmic
*SpHSP19.8*	Sp08G006210.1	08	5,837,234–5,840,830	177	19.81869	6.62	51.03	Cytoplasmic
*SpHSP20.3A*	Sp12G000650.1	12	703,223–704,282	184	20.31725	9.06	49.37	None
*SpHSP20.3B*	Sp16G008980.1	16	7,399,445–7,400,631	182	20.34325	9.1	43.73	None
*SpHSP21.5*	Sp07G022390.1	07	32,111,598–32,112,257	195	21.45444	8.83	48.78	Mitochondrial
*SpHSP23.2A*	Sp05G017700.1	05	29,709,434–29,710,054	206	23.1906	6.14	53.52	Cytoplasmic
*SpHSP23.2B*	Sp15G038330.1	15	48,982,836–48,984,657	207	23.20383	8.75	35.16	Cytoplasmic
*SpHSP23.4*	Sp10G016790.1	10	28,990,770–28,991,390	206	23.42879	6.03	51.74	Cytoplasmic
*SpHSP23.5*	Sp11G021010.1	11	30,127,432–30,129,736	206	23.53936	5.54	43.32	Cytoplasmic
*SpHSP23.7*	Sp03G019560.1	03	29,437,763–29,441,016	207	23.73253	5.19	46.64	Cytoplasmic
*SpHSP23.9*	Sp08G022710.1	08	31,018,189–31,018,830	213	23.87339	9.25	34.87	Cytoplasmic
*SpHSP24.4*	Sp01G020080.1	01	26,758,772–26,759,869	222	24.41663	9.02	46.29	Chloroplast
*SpHSP24.7*	Sp07G026110.1	07	35,681,144–35,682,720	223	24.69794	7.88	48.57	Chloroplast
*SpHSP26.0*	Sp08G006230.1	08	5,848,645–5,849,376	243	25.98482	6.01	57.56	Cytoplasmic
*SpHSP26.7*	Sp13G008820.1	13	7,558,891–7,560,398	240	26.73056	9.15	43.59	None
*SpHSP26.9*	Sp09G019280.1	09	21,477,023–21,479,489	241	26.92018	6.98	60.49	Mitochondrial
*SpHSP27.0*	Sp15G004360.1	15	3,382,832–3,383,631	253	27.0497	5.78	63.12	Cytoplasmic
*SpHSP27.2*	Sp17G019110.1	17	25,825,335–25,826,764	243	27.16939	7.77	52.43	Mitochondrial
*SpHSP30.6*	Sp08G006220.1	08	5,841,762–5,842,619	285	30.63445	5.42	51.72	Cytoplasmic
*SpHSP31.5*	Sp06G005610.1	06	8,090,079–8,091,509	282	31.53698	9.39	71.26	Chloroplast

## Data Availability

Data are contained within the article or Appendix A.

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
