# Peer review of "Genome-Wide Identification and Expression Profiling of Heat Shock Protein 20 Gene Family in Sorbus pohuashanensis (Hance) Hedl under Abiotic Stress"

_genes, 2022, doi:10.3390/genes13122241_

Round 1

Reviewer 1 Report

This article presented Genome-Wide Identification and Expression Profiling of Heat Shock Protein 20 Gene Family in Sorbus pohuashanensis (Hance) Hedl Under Abiotic Stress in a good way. However, there is a major deficiency, which is lack of molecular characterization.

Present specific results of expression pattern.

Line 3 introduction should be cited with recent studies. https://doi.org/10.1016/j.pmpp.2021.101639, https://doi.org/10.3390/ijms22179175, https://doi.org/10.3390/genes13101699,

Also check reference and citation style throughout the MS.

Consecutive references should be write as (7-10).

These lines must be cited “genome sequencing have been used to analyze the

molecular regulatory mechanisms involved in response to heat stress in S. pohuashanensis

Section 2.3 write how the Phylogenetic tree was visualized.

Section 2.7 should be cited with relevant study. https://doi.org/10.1007/s10725-021-00785-7,

Discussion needs further improvement by comparing with recent studies.

Conclusion is well justified. The authors should discuss some points for the future studies molecular level studies are required to know about the involved mechanism and improvement of abiotic stress.

Reviewer 2 Report

It's a very well done manuscript. There are only a few minor grammatical errors. I indicated it by highlighting on the pdf and writing notes. I think it is appropriate to print it after these minor changes.

Kind regards.
